# Unravelling Morphological and Topological Energy Contributions of Metal Nanoparticles

**DOI:** 10.3390/nano12010017

**Published:** 2021-12-22

**Authors:** Lorena Vega, Francesc Viñes, Konstantin M. Neyman

**Affiliations:** 1Departament de Ciència de Materials i Química Física, Universitat de Barcelona, c/Martí i Franquès 1-11, 08028 Barcelona, Spain; lvega@ub.edu (L.V.); konstantin.neyman@icrea.cat (K.M.N.); 2Institut de Química Teòrica i Computacional (IQTCUB), Universitat de Barcelona, c/Martí i Franquès 1-11, 08028 Barcelona, Spain; 3Institució Catalana de Recerca i Estudis Avançats (ICREA), Pg. Lluís Companys 23, 08010 Barcelona, Spain

**Keywords:** cohesive energy, density functional calculations, metal nanoparticles, size and shape dependence, multilinear regression, Wulff structures, surface energies, coordination numbers

## Abstract

Metal nanoparticles (NPs) are ubiquitous in many fields, from nanotechnology to heterogeneous catalysis, with properties differing from those of single-crystal surfaces and bulks. A key aspect is the size-dependent evolution of NP properties toward the bulk limit, including the adoption of different NP shapes, which may bias the NP stability based on the NP size. Herein, the stability of different Pd*_n_* NPs (*n* = 10–1504 atoms) considering a myriad of shapes is investigated by first-principles energy optimisation, leading to the determination that icosahedron shapes are the most stable up to a size of ca. 4 nm. In NPs larger than that size, truncated octahedron shapes become more stable, yet a presence of larger {001} facets than the Wulff construction is forecasted due to their increased stability, compared with (001) single-crystal surfaces, and the lower stability of {111} facets, compared with (111) single-crystal surfaces. The NP cohesive energy breakdown in terms of coordination numbers is found to be an excellent quantitative tool of the stability assessment, with mean absolute errors of solely 0.01 eV·atom^−1^, while a geometry breakdown allows only for a qualitative stability screening.

## 1. Introduction

Over the last few decades, nanomaterials have become ubiquitous in various industrial and/or technological applications, including, e.g., energy storage [1], antimicrobial agents [2], selective release of drugs [3], heterogeneous catalysts [4], etc. In the latter field, nanostructuring, i.e., the use of metallic nanoparticles (NPs), has become an ubiquitous way of improving the efficiency of a catalyst while reducing the employed amount of it, a key point when using precious and expensive late transition metals, regular active phases in a large number of catalytic processes [5]. Paramount examples are, e.g., the renowned increased catalytic activity of Au NPs in the oxidation of CO when the NPs were supported on TiO_2_ [6], or the increase in selectivity toward pyrrolidine, compared with n-butylamine when reducing the size of the employed Pt NPs in the catalysis of pyrrole hydrogenation [7].

From an atomic-level point of view, the catalytic activity of a given metal NP is the result of exposing special sites involving low-coordinated atoms, such as those located at NP corners, edges, or facets, and their peculiar electronic structure [4,5]. For instance, low-coordinated gold atoms have been identified as key sites in the catalytic dissociation of H_2_ molecules [8]. In addition, particle shape often varies with particle size [9], which can, in turn, affect the number of exposed low-coordinated sites, ultimately biasing the overall NP catalytic activity. Consequently, size- and shape-dependent properties of metal NPs make their catalytic properties tuneable [10,11,12,13].

It is for this reason that unravelling the morphology and topology of metal NPs becomes crucial. Experimentally, different studies tackled this issue using metallic NPs deposited on a variety of supports [14,15,16,17]. Given the technical difficulties in experimentally controlling size and shape, usually, obtaining a distribution of them, plus the inherent averaged experimental analysis, many studies relied on computational simulations to gain the necessary atomistic insight, allowing even for the NP analysis in the absence of support, and therefore, weighting its effect [18,19,20,21,22], compared with models accounting for the support [23,24]. However, such computational studies are not exempt of difficulties; for instance, when dealing with NP models of transition metals within the so-called scalable regime, a size limit of ~100 atoms from which the metal NPs properties scale linearly with size [4], their first-principles computational simulation is at the frontier of the computational power of modern computing cluster architectures and the capabilities of standard codes.

However, the rise of high-performance parallel supercomputers nowadays, coupled with massively parallelised modern quantum computing codes, allows a leap forward in the explicit treatment of NPs containing from a few hundred to a few thousand atoms. This has been already successfully demonstrated for oxide semiconductors (TiO_2_ and ZnO), unfolding a rich diversity of structures in the nanoregion [25,26]. Such calculations were performed employing density functional theory (DFT), the regular working horse when studying metal NPs, yielding accurate results at a reasonable computing time. For transition metals (TMs), generalised gradient approximation (GGA) exchange-correlation functionals such as the Perdew–Burke–Ernzerhof (PBE) [27] have been found to be among the best describing TMs bulk and surface properties [28,29]. However, as long as NP modelling is concerned, even if periodic surface models are extremely useful [28], a duly simulation requires the employment of well-shaped isolated NPs.

This was shown in the seminal work of Yudanov et al., which was exemplified on Pd NPs [30], exploiting the NPs point group symmetries, and later applied to Pd and other late TMs using a periodic code and employing plane waves as basis set [31,32]. This allowed the growing use of such models to study heterogeneous catalysts and nanotechnology devices, profiting from the aforementioned description scalable towards the bulk limit or converged with size [4,33], and the wise combination of NP with periodic slab models to present a rather complete description of larger NPs [34,35].

Here, we move forward to exploit high-performance parallel supercomputing facilities combined with a highly parallelised computational code using numeric atom-centred orbitals (NAOs) to push the limit of metal NPs description, allowing for a full shape analysis as a matter of size and permitting breakdown of the NPs energies either in terms of atomic contributions related to their coordination number (CN) or as NP geometric factors; both analyses allow predicting the energy of any NP independently of its size or shape. To this end, Pd NPs, a common playground of previous analyses [30,31] are inspected, and a nanomaterial of catalytic interest, e.g., nanoengineered Pd NPs, are used in nitrite reduction [36], or as catalysts for Suzuki cross-coupling reactions [37].

## 2. Computational Details

Pd*_n_* NPs with *n* ranging from 10 to 1504 atoms, this is, from the sub nm size up to a size of ~4 nm, have been modelled for a wide variety of shapes. Specifically, in addition to the truncated octahedron (To) Wulff construction shape for Pd_n_ minimising the overall NP surface tension [38], other experimentally reported [39,40,41,42] shapes were considered that retain the bulk Pd face-centred cubic (*fcc*) arrangement such as cube (C), truncated cube (Tc), octahedron (Oh), cuboctahedron (Ch), spheres (S), tetrahedron (Th), and decahedron (Dh) shapes, plus icosahedrons (Ih) with a distorted bulk Pd structure (Figure 1). All the shapes have been cut from *fcc* Pd bulk, except for icosahedron shapes, and spheres; the latter are built from *fcc* Pd bulk but define an NP sphere radius to cut off outer Pd atoms.

Two main properties of the NPs were analysed: one geometrical, that is, the average interatomic distance between neighbouring Pd atoms, δ(Pd–Pd), and one energetic, in this case, the mean atomic cohesive energy, *E_coh_*. Such features can be analysed depending on the NPs size, accounted either by *n*^−1/3^, that is, following the spherical cluster approximation [43], or by the average coordination number, CN_av_, to examine the size-dependent evolution of NPs properties towards the bulk limit [4,31,32,44,45], reached at *n*^−1/3^ = 0 and CN_av_ = 12 of bulk *fcc* Pd. In this way, 94 NPs with different shapes were modelled; the list of shapes is presented in Appendix A. These NPs account for a large diversity of sites with different CNs, allowing for the energy breakdown based on either topological contributions [44] or geometry components, as carried out earlier to isolate energies of steps on CeO_2_ islands [46] and energies of a row on Cu surfaces [47].

The total energies of the locally optimised NPs were computed using non-spin-polarised DFT calculations employing the PBE exchange-correlation functional [27], as implemented within the all-electron full-potential Fritz–Haber institute ab initio molecular simulations (FHI-AIMS) package [48]. There, Kohn–Sham orbitals were expanded using numeric atomic orbitals [49,50], hierarchically constructed by adding functions to a minimal basis set to yield an accuracy of the total energy at the meV level. The calculations were performed employing the first light-tier basis set, which includes all the most important basic functions. Relativistic effects were considered through the scaled zeroth-order regular approximation (ZORA) [50,51].

To achieve the self-consistency of the electron density optimisation, two criteria were imposed: the differences between consecutive steps of the total energy and atomic forces set to 1·10^−6^ eV and 1·10^−4^ eV·Å^−1^, respectively. Furthermore, a Gaussian smearing parameter of 0.3 eV was used to speed up convergence. Geometrical optimisations were performed using the Broden–Fletcher–Goldfarb–Shanno (BFGS) algorithm [52,53], and equilibrium geometries were found once all atomic forces were smaller than 1·10^−2^ eV·Å^−1^.

## 3. Results and Discussion

Once the Pd*_n_* NPs of different shapes, shown in Figure 1, were optimised, the size-dependent evolution of the NPs properties towards the bulk limit was tackled. Notably, the different shapes provide different surface features, i.e., different types of exposed facets, edges, and corners, which are later used to break down the NPs energies. Following previous studies, the first evaluated was δ(Pd-Pd) distance vs. *n*^−1/3^ (Figure 2). The average minimum interatomic distances δ(Pd-Pd) calculated for each NP are listed in Appendix A. A linear regression model for each shape family was obtained, with intercepts, slopes, and regression coefficients listed in Appendix A. The linear trends reveal structural information, and as in all cases, δ(Pd–Pd) increases with NP size approaching the bulk limit, here estimated to be 2.79 Å, succinctly implying NPs become more shrunk when reducing their size. The only outliers of this trend are decahedron NPs, which feature a slightly larger extrapolated bulk limit of 2.82 Å, although this extrapolation could be biased by the reduced number of NPs used in the linear regression.

Notably, similar cuboctahedron, octahedron, truncated octahedron, and truncated cube shapes evolve similarly with size, also found for spherical NPs (Figure 2). An icosahedron is a shape that features the longest distances, which becomes particularly evident for smaller NPs. Nevertheless, the maximum elongation of 0.03 Å is found for Ih Pd_55_, compared with Pd_55_, as a result of distorting the inner core NP *fcc* crystal structure at icosahedron shapes. Conversely, cube, tetrahedron, and decahedron shapes are the ones with the shortest δ(Pd–Pd) distances, again especially for small NPs, due to the surface strain of their peculiar edges.

In order to assess the stability of the shape at different NP sizes, a linear regression model for each shape was performed, shown in Figure 2, where the *E_coh_* values of each NP listed in Appendix A were fitted respective to *n*^−1/3^, (see regression parameters in Appendix A). As one can readily observe, the icosahedron is the most stable shape for smaller NPs, understandable as all the exposed surfaces are (111)-like, which is the most stable surface for Pd and other *fcc* TMs in general. Indeed, the icosahedron shape can be the most stable as long as the reduction in surface tension energy compensates for the inner core deformation. The present results are in line with data of previous studies reporting icosahedron shapes as the most stable for small NPs [54,55].

However, the icosahedron shape ceases to be the preferred one at larger NPs sizes; from *n*^−1/3^ = 0.053, i.e., ca. *n* = 6530, and NPs with a diameter Ø~7 nm, the most stable shape is the sphere, surprisingly different from the expected truncated octahedron shape derived from the Wulff construction [38] and thus behaving differently from other similar TMs, such as Pt [56]. A possible reason for this peculiar behaviour is that Wulff construction, in its mathematical shaping ansatz, accounts for neither edge nor corner energies. Aside from this fact, as already mentioned, spherical NPs were shaped by cutting off Pd atoms beyond a defined radius measured from the centre of an *fcc* bulk; this actually yields, for very small spherical NPs, a truncated octahedron-like shape.

Furthermore, within the truncated octahedrons family, it is possible to differentiate subgroups, depending on the degree of exposition {001} and {111} facets. Plotting subfamilies, the crossing points, and stabilities vary (see linear regression coefficients in Appendix A). As shown in Figure 3, the icosahedron shape is most stable up to *n*^−1/3^ = 0.08, i.e., ca. *n* = 1500, and NPs with a diameter of Ø~4 nm. At larger sizes, the truncated octahedron shape with a large exposure of {001} facets is preferred; indeed, close to a cuboctahedral shape but still quite more stable than the latter and the octahedron shape (Appendix A). From the analysis of different truncated octahedron subfamilies, it seems clear that the stability is reached upon exposing larger {001} facets. Still, one has to be cautious with such adjustments as for some cases the number of points in the linear regression is limited. Nevertheless, the present assessment indicates that the Wulff shape is an appropriately educated guess for large NPs, but this simplification misrepresents, as the present explicit calculations reveal, a larger {001} facet exposure than that resulting from the Wulff approach [28,38].

Small energy difference for different shapes, of ca. 0.02 eV/atom in the nm size region, implies that Pd NPs are malleable and, therefore, could easily adopt different shapes, as found experimentally [39,40,41,42]. Such shape modification can also be facilitated by the medium, in the sense that the released energy due to the adsorption of certain species on Pd NPs may compensate for the energy cost required to change the shape. Additionally, different shapes can be induced by the support, as Engel et al. pointed out for Au NPs on MgO, Carbon, and CeO_2_ [57].

Going beyond the mere size dependency of δ(Pd–Pd) and *E_coh_* of Pd NPs, the obtained data for a wide variety of NPs and shapes representing altogether 94 independent cases allow for an energy breakdown in terms of geometrical parameters, as well as atomic coordination environment, which could ultimately enable predicting the energy of unexplored larger or different shape Pd NPs. This is tackled in the following focusing on either the atomic CN or geometrical features such as the number of vertexes, length of edges, surface areas, and NP volume.

For both approaches, a multilinear regression model was applied, a simple breakdown process, based on which understandable conclusions can be drawn. More than simply the description, creating predictive tools was also envisaged. Thus, following machine learning protocols, one would ideally split the data into training and test sets. However, since the number of data was limited, that could bias the model depending on the selection of data for each set. Therefore, the regression was cross-validated by a shuffle split [58]. Briefly, the full set of data was randomly split *m* times in training and test sets, and each split was fitted and evaluated. Here, *m* = 100, and for each random split, ¾ of the data was assigned to training and ¼ to testing, as shown in Appendix A. The henceforth discussed regression coefficients and errors were thus averaged over 100 fittings.

The first proposed model was to decompose the NP energy, employing the atomic *E_coh_*, as a function of the number of atoms and the *CN* of each type [44]. Thus,
(1)$$E_{coh}=\overset{CN}{\underset{i=1}{\sum }}\varepsilon_{i}\chi_{i}+C$$
where $\varepsilon_{i}$ is the energy contribution to the *E_coh_* of the atoms with a given *i CN*, and $\chi_{i}$ is the fraction of Pd atoms with the *i CN* with respect to the total number *n* of Pd atoms (Appendix A). It is worth noting that an independent term was considered, whereas the terms $\varepsilon_{12}$ and $\chi_{12}$ were not considered in the multilinear regression, since $\chi_{12}$ was already correlated to the other atomic fractions. Further, *CN* = 1 and 2 terms were not featured in any of the studied NPs and, consequently, disregarded as well. Moreover, icosahedron shapes were not initially considered since, different from the other studied shapes, they have distorted core *fcc* structures. Within such a treatment, the independent *C* term is the *fcc* Pd bulk cohesive energy and the $\varepsilon_{i}$ coefficients describe an energy destabilisation contribution with respect to the Pd *fcc* bulk environment. Following this procedure, the obtained mean $\varepsilon_{i}$ coefficients are as follows:(2)$$E_{coh}=-3.72+1.95\chi_{3}+1.88\chi_{4}+1.47\chi_{5}+1.29\chi_{6}+1.01\chi_{7}+0.66\chi_{8}+0.52\chi_{9}+0.14\chi_{10}+0.04\chi_{11}$$

All coefficients decrease the atomic cohesive energy, and the lower the *CN* is, the more the energy is decreased. Figure 4 evidences linearity of coefficient destabilisation with respect to *CN*, a finding to be expected for similar TMs. Notably, very similar coefficients and a similar evolution of them with changing *CN* are achieved when accounting for the icosahedron shape, as shown in Equation (S1) and Appendix A.

Exceeding the just mentioned basic adjustment, the mean test errors of the gained equation show an excellent agreement to the created model, with a regression coefficient of *R*^2^ = 0.995, and, consequently, an excellent associated mean absolute error (MAE) of solely 0.011 eV/atom. When accounting for icosahedron shapes, the *R*^2^ becomes just slightly reduced, to *R*^2^ = 0.972, and a twice larger MAE of 0.022 eV/atom is found. These results suggest that such an energy breakdown is suited for a quantitative analysis even when mixing different bulk crystal structures. Although the adjustment is visually quite accurate for *fcc*-based NPs (Figure 5), it clearly shows icosahedron shapes as outliers, as revealed by Appendix A.

To assess the prediction capacity of the CN breakdown, a learning curve of the cross-validation (CV) analysis for training and test sets is shown in Figure 5 and Appendix A, increasing the number of training set points while keeping the training ratio. As expected, a small MAE below 0.02 eV/atom is achieved already for very small samples in the training set. On the CV, an MAE~0.01 eV/atom is achievable using a reasonably small number of ca. 40 training samples, but fewer samples yield larger errors, given the low quantity of fitting data, compared with the nine $\chi_{i}$ variables, where at least the same number of data as variables is needed to solve the system equations. Thus, underfitting appears to be at the origin of the larger errors.

As far as the geometric analysis is concerned, *E_coh_* is fitted without defining any intercept for the topological features including the number of corners (*C*), the total longitude of edges (*L*), the total exposed area (*A*), and the NP volume (*V*), normalised by the number of Pd atoms, *n*, to have comparative values across sizes. Thus,
(3)$$E_{coh}=\varepsilon_{C}\frac{C}{n}+\varepsilon_{L}\frac{L}{n}+\varepsilon_{A}\frac{A}{n}+\varepsilon_{V}\frac{V}{n}$$
where $\varepsilon_{C}$, $\varepsilon_{L}$, $\varepsilon_{A}$
, and $\varepsilon_{V}$ are the contributions to the atomic cohesive energy of the NP corners, edges, surfaces, and volume, respectively. In this case, only regular shapes were considered, excluding the sphere, truncated octahedron, and truncated cube shapes from the analysis. Thus, the dataset was reduced to 40 cases for only four fitting parameters, instead of the nine parameters used for the CN-based energy breakdown. Applying the same CV as explained before for the *E_coh_* breakdown as a function of the atomic CN and considering the same *m* = 100 and percentages of training (75%) and test (25%) sets, the obtained mean coefficients are as follows:(4)$$E_{coh}=0.54\frac{C}{n}-0.08\frac{L}{n}-0.29\frac{A}{n}-0.23\frac{V}{n}$$

These results clearly reveal that corners are the only topologic points detrimental to the atomic cohesive energy by 0.54 eV, due to their very low coordination. In this sense, NP edges slightly contribute to stabilising the shape by −0.08 eV, and even more the surfaces, with the contribution of −0.29 eV, thus in line with the increased coordination. Volume substantially contributes to the stabilisation by −0.23 eV but, curiously, less than the surface. Such a result has to be taken with a grain of salt, since the NP volume is the topological feature actually contributing more to the NP cohesion, due to its highest coordination. This is because the NP volume is extensively larger than its exposed surface, particularly true for large NPs, as indicated by the number of atoms with CN 12 in Appendix A and topological data in Appendix A.

The calculated and predicted model *E_coh_* values are compared in Figure 6. Clearly, the geometry breakdown provides a looser description, compared with the CN breakdown, with the data rather dispersed. This is translated into a poorer regression coefficient of *R*^2^ = 0.763 and a significantly higher MAE of 0.110 eV/atom. Even if the learning curve in Figure 6 reveals that the training set can reach rather accurate levels with a reduced number of samples, the CV score needs at least 15 points for the MAE to converge to 0.11 eV/atom. This refrains from quantitative using this energy breakdown method, although it seems useful for rapid, qualitative assessments.

The accuracy of this method is biased by the small number of variables, which often mix different situations. For instance, all exposed surfaces are treated equally, e.g., considering (001) and (111) types of facets the same, even if their surface energy is different [28]. The same occurs for edges, as there are different types, depending on via which side facets connect, that is, it is expectable that separating edges (001) and (111) facets have different energy than those between (111) facets. Lastly, different kinds of corners are present, with different CNs and spatial positions of the neighbouring atoms.

To assess this effect, we decomposed, for the cube and octahedron families, the total energy of the samples as a function of bulk *fcc* Pd energies, plus the contributions to the energy corresponding to the surface, edge, and corner energies, following a procedure earlier employed to obtain step and row energies [46,47]. According to the procedure, the cubic and octahedron NPs energies can be decomposed as
(5)$$E=n\varepsilon_{bulk}+\gamma A+\beta L+\xi C$$
where $\varepsilon_{bulk}$ is the energy of a single Pd atom in the bulk Pd *fcc* structure, and γ, β, and ξ are surface, edge, and point energies, respectively. Notably, for cubic NPs, the whole exposed area *A* belongs to (001) type of surfaces, and so γ corresponds to the (001) surface energy, while *L* longitudes belong to edges between (001) facets and, therefore, define the edge energy, β. Lastly, the corner energies, ξ, are for corners having vicinal (001) facets. Likewise, one can decompose the energy of octahedron NPs but relating to the (111) facets only.

The obtained surface, edge, and corner energies are displayed in Table 1. From cubes and octahedrons, for (001) and (111) facets, energy values of 0.09 and 0.08 eV·Å^2^ are achieved, that is, values of 1.41 and 1.24 J·m^−2^, respectively. These values somewhat differ from the extended surfaces estimations using slab models of 1.5 and 1.14 J·m^−2^. Thus, on such Pd NPs, (001) surfaces become somewhat more stable, by 0.09 J·m^−2^, while (111) surfaces become less stable, by 0.10 J·m^−2^. This difference has its origins in the reduction in δ(Pd–Pd) distance, and how this affects the particular energy of such facets, as well as in the stabilisation or destabilisation of the Pd atoms located at the facets’ boundary regions. Regardless of this fact, these data explain the truncated octahedron preferential shape, shown in Figure 3, as the (001) surfaces are abundantly exposed in this shape.

Interestingly, this change of shape does not compromise edge energy, as it is the same value of 0.27 eV·Å^−1^ for edges connecting either (001) facets or (111) facets. The point energy of corners differs substantially, from 0.07 eV for cubes, to −0.50 eV for octahedrons; however, these contributions barely affect the overall surface tension, as their number are quite small, compared with the exposed facets area and edges lengths, particularly true for large NPs (Appendix A).

## 4. Conclusions

The present study addressed explicitly optimised DFT Pd*_n_* clusters and NPs, including a vast myriad of shapes and reaching unprecedented sizes of *n*~1500 atoms and diameter Ø~4 nm. The size evolution of the mean neighbouring Pd–Pd distances, δ(Pd–Pd), and the atomic cohesive energy, *E_coh_*, towards the bulk limit were shown to be linear with respect to *n*^−1/3^ for all considered NP shapes. From the analysis of NP shape dependence on their size, the icosahedral shape with the bulk structure distorted vs. *fcc* resulted in the most stable for NPs, with up to 1500 Pd atoms. In NPs with larger sizes than this, the truncated octahedron shape with a large exposure of (001) facets appeared to become the most stable. The Wulff construction model was partially followed but with a presence of larger (001) facets. The latter was shown here to be due to the stronger stabilisation of (001) facets and an equal destabilisation of (111) facets.

The large number of studied structures, i.e., 94, and variety of shapes, i.e., 9 different families, allowed decomposing *E_coh_* in terms of atomic contributions grouped by CN, as well as geometric contributions accounting for the number of corners, edges lengths, facets areas, and NP volumes. The energy breakdown based on CNs was found to be quantitative, with an MAE of solely 0.01 eV·atom^−1^, for NPs with *fcc* arrangement of Pd bulk. Including icosahedrons with a different arrangement of inner Pd atoms only slightly increased the MAE to 0.02 eV·atom^−1^. The breakdown revealed that the smaller the CN is, the larger the destabilisation from the bulk cohesive energy is.

The breakdown employing geometric terms, performed on a subset of 40 NPs with a clearly defined geometry, featured a poorer accuracy, with an MAE of 0.11 eV·atom^−1^. Nevertheless, this is sufficient for a qualitative assessment, for instance, revealing that corner points were the only destabilising geometric feature. The reduced accuracy of this model is related to accounting for all exposed facets, edges, and corners with the same energetic contributions, although the present DFT data analysis revealed that they may significantly differ from each other.

## Figures and Tables

**Figure 1 nanomaterials-12-00017-f001:**
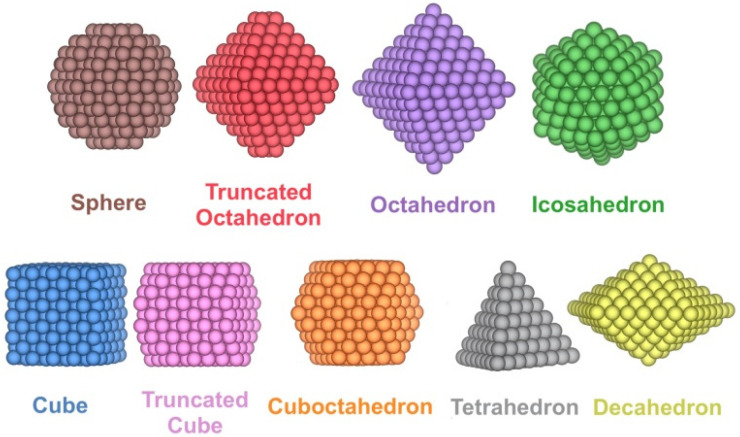
Examples of the different shapes studied for Pd_n_ NPs.

**Figure 2 nanomaterials-12-00017-f002:**
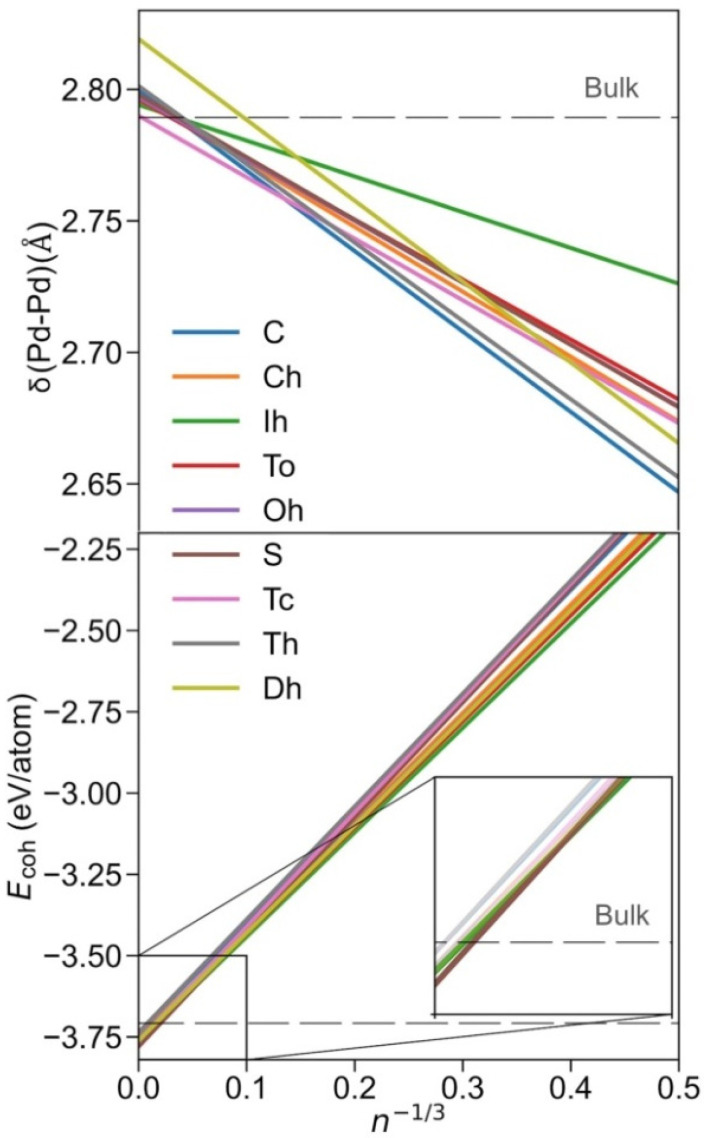
Top panel: evolution of δ(Pd–Pd) with size, here estimated by *n*^−1/3^, where *n* is the number of Pd atoms of the NP. For better readability of trends, only linear regressions are shown. Bottom panel: evolution of the cohesive energy, *E**_coh_*, with *n*^−1/3^. Colour coding is as in Figure 1.

**Figure 3 nanomaterials-12-00017-f003:**
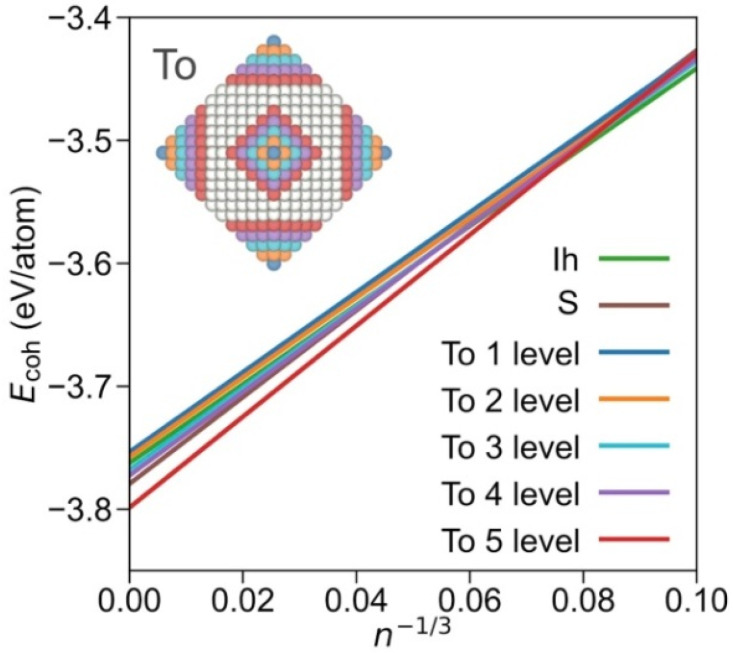
Comparison of most stable Icosahedron and sphere shapes and truncated octahedron (To) shapes, split by each level of {001} cuts departing from octahedron shapes. Inset NP image illustrates the removed atoms from each level from an octahedron NP. The fitting lines obtained by the resulting truncated octahedron groups are shown in the colours of the removed atoms. Sphere and icosahedron colour coding as in the signalled linear adjustments.

**Figure 4 nanomaterials-12-00017-f004:**
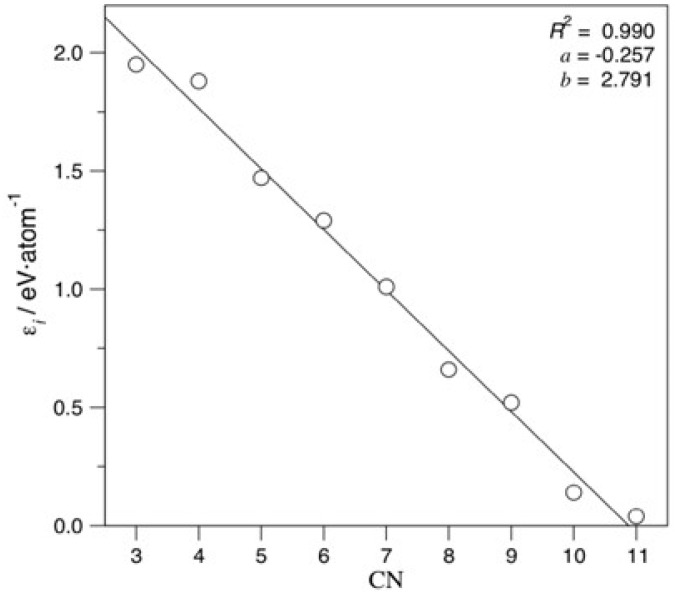
Linear adjustment coefficients $\varepsilon_{i}$ vs. CN. Regression coefficient, *R*^2^, slope, a, and intercept, b, were also specified.

**Figure 5 nanomaterials-12-00017-f005:**
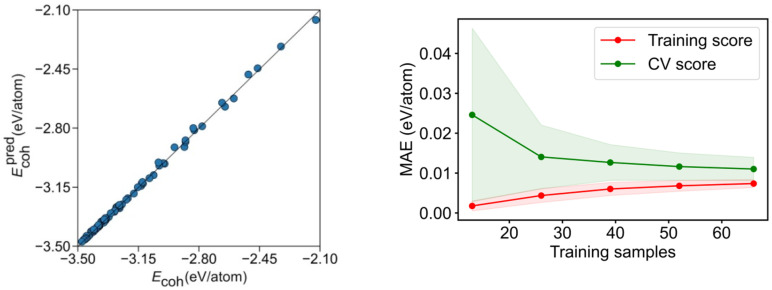
Left panel: comparison of calculated *E_coh_* vs. the predicted *E_coh_*, $E_{coh}^{pred}$, for the CN breakdown. Right panel: MAE learning curve for the training and CV scores. The coloured areas represent the standard variation limits as a result of the 100 different fittings.

**Figure 6 nanomaterials-12-00017-f006:**
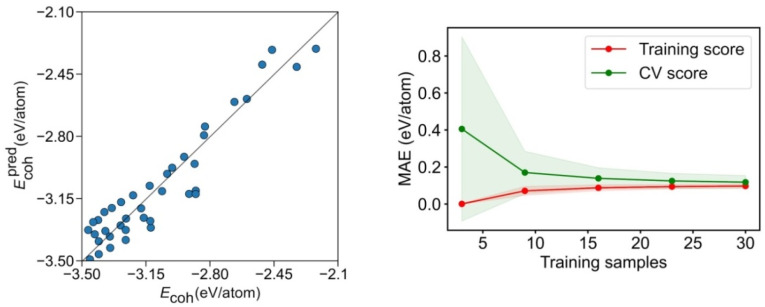
Left panel: comparison of calculated E*_coh_* vs. the predicted E*_coh_*, $E_{coh}^{pred}$, for the geometry breakdown. Right panel: MAE learning curve for the training and CV scores. The coloured areas represent the standard variation limits as a result of the 100 different fittings.

**Table 1 nanomaterials-12-00017-t001:** Surface, γ, edge, β, and point energies, given in eV·Å^−2^, eV·Å^−1^, and eV, respectively, estimated for cubic and octahedral Pd NPs using Equation (5).

NPs	*γ* _(001)_	*γ* _(111)_	*β* _(001)_	*β* _(111)_	*ξ*
C	0.09	—	0.27	—	0.07
Oh	—	0.08	—	0.27	−0.5

## Data Availability

The data presented in this study are available on request from the Informed Consent Statement: corresponding author.

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
