# Peer review of "Unravelling Morphological and Topological Energy Contributions of Metal Nanoparticles"

_nanomaterials, 2021, doi:10.3390/nano12010017_

Round 1

Reviewer 1 Report

This paper describes computational studies about the energetic of Pd nanoparticles with different geometries.

The paper is well organized, the results are clearly exposed and the conclusions are convincing.

In my opinion, the paper describes better methodological aspects (algorithms, equations) rather than chemical-physical results.

As a matter of facts, the differences between alternative geometries are quite small, and the author themselves admit that the experimental NP can be affected by many other factors (such as surface adsorption, defects, irregular shapes).

An additional medium effect, which can be outlined (line 200), is the interaction with the support.

The results are limited to only one element (Pd), and fcc packing (which is adopted in the bulk metal). I suggest to consider (at least) one alternative hcp, which may be of comparable energy in this nanometric regime.

Author Response

Please, see attached document.

Reviewer 2 Report

Authors studied the size-dependent evolution of metal nanoparticle properties by using first-principles calculations. Different aspects of evolution are studied, including different nanoparticle shapes, stability, cohesive energy, interatomic distance and so on. The authors concluded that Icosahedron shapes are the most stable shape for Pdn nanoparticles below the size of ca. 4nm, while Truncated Octahedron shapes are more stable above. The result also shows that the nanoparticle cohesive energy breakdown in terms of coordination numbers is a better quantitative tool than the geometry breakdown. The manuscript is well written and contains a set of good data. I will support its publication after my questions can be answered.

1) For the nanoparticle cohesive energy breakdown in Fig. 4 and 5, the model does not consider any nanoparticle shapes. But in Fig. S3 and S4 in the Supplementary Material, results are shown by considering Icosahedrons. Results show slightly different. I’m confused that what’s the difference between the two models. In other words, when considering a certain special shape such as Icosahedron, which parameter should be modified? Why this parameter and how to modify it with different shapes? Besides, why the model in Fig. 4 and 5 is universal with different shapes?

2) Same question as (2) to the geometry breakdown.

Minor issue:

In figure 3, Truncated Octahedron shapes split by each level of {001} cuts from Octahedron shapes are shown in the inset nanoparticles image. The atoms are illustrated by different colors. I suppose that different colors in the atoms correspond to different cut levels, which also correspond to the color coding as in the signalled linear adjustments. It would be better if the author could describe it in the figure caption.

Author Response

Please, see attached document.
